# Comparison of Immune Responses between Inactivated and mRNA SARS-CoV-2 Vaccines Used for a Booster Dose in Mice

**DOI:** 10.3390/v15061351

**Published:** 2023-06-11

**Authors:** Ning Luan, Han Cao, Yunfei Wang, Kangyang Lin, Jingping Hu, Cunbao Liu

**Affiliations:** Institute of Medical Biology, Chinese Academy of Medical Sciences and Peking Union Medical College, Kunming 650118, China; luanning@imbcams.com.cn (N.L.); caohan@imbcams.com.cn (H.C.); wangyf@imbcams.com.cn (Y.W.); linky6679@163.com (K.L.); hujingping430@126.com (J.H.)

**Keywords:** SARS-CoV-2, VOCs, mRNA vaccines, booster vaccines

## Abstract

A large amount of real-world data suggests that the emergence of variants of concern (VOCs) has brought new challenges to the fight against SARS-CoV-2 because the immune protection elicited by the existing coronavirus disease 2019 (COVID-19) vaccines was weakened. In response to the VOCs, it is necessary to advocate for the administration of booster vaccine doses to extend the effectiveness of vaccines and enhance neutralization titers. In this study, the immune effects of mRNA vaccines based on the WT (prototypic strain) and omicron (B1.1.529) strains for use as booster vaccines were investigated in mice. It was determined that with two-dose inactivated vaccine priming, boosting with mRNA vaccines could elevate IgG titers, enhance cell-mediated immunity, and provide immune protection against the corresponding variants, but cross-protection against distinct strains was inferior. This study comprehensively describes the differences in the mice boosted with mRNA vaccines based on the WT strain and the omicron strain, a harmful VOC that has resulted in a sharp rise in the number of infections, and reveals the most efficacious vaccination strategy against omicron and future SARS-CoV-2 variants.

## 1. Introduction

Coronavirus disease 19 (COVID-19), caused by severe acute respiratory syndrome coronavirus 2 (SARS-CoV-2), rapidly caused a pandemic worldwide, resulting in an exponential increase in the number of cases. A total of 13 vaccines in various forms, such as inactivated vaccines, messenger RNA (mRNA) vaccines, subunit vaccines, and adenovirus-based vaccines, have been approved by the World Health Organization (WHO) for emergency use to defend against SARS-CoV-2 infection. The robust production technology and high safety have made alum-adjuvanted inactivated SARS-CoV-2 vaccines, such as CoronaVac (Sinovac Life Sciences) and KconVac (Shenzhen Kangtai/Beijing Minhai Biotechnology), the most used since the beginning of the pandemic [1]. Formulated with lipid-based nanoparticles (LNPs), mRNA vaccines can induce the intracellular production of antigens and mobilize the innate immune system of organisms by inducing both the production of high neutralization titers and robust cellular immunity [2,3]. BNT162b2 (BioNTech/Fosun Pharma/Pfizer) and mRNA-1273 (ModernaTx) have been approved by the US Food and Drug Administration (FDA) against the prototype of SARS-CoV-2 [4,5]. To date, vaccinations have saved more than 20 million lives from COVID-19 and provided protection for critically ill patients [6].

However, regardless of which form of vaccine was received, it appears that the protection afforded against viral infection diminishes after 3–6 months. Moreover, SARS-CoV-2 mutates rapidly during infection, and several variants of concern (VOCs) have emerged and resulted in the deaths of millions of people [7]. In the context of VOCs, the efficacy of neutralizing antibodies (nAbs), whose production is induced by vaccination against the prototype of SARS-CoV-2, has waned significantly. Several studies have demonstrated that as VOCs can partially evade the immunity provided by the original vaccines, the risk of secondary infection remains despite full-schedule vaccination and booster vaccination strategies are being considered to improve vaccine efficacy against constantly emerging VOCs [8].

In our previous study, we compared the immunization effects between 0–28–84 days and 0–28–168 days after vaccination induced by the inactivated SARS-CoV-2 vaccine, which is based on the prototypic strain [9]. It was demonstrated that protection at 2 months (group 0–28–84 days) was superior to that at 5 months (group 0–28–168 days), corresponding to the interval between the second vaccination and the third boosting. Thus, 0–28–84 days was set as the best choice for the administration of the third booster vaccination. Moreover, the immune effect of inactivated booster vaccines and different isotype mRNA booster vaccines in the context of VOCs should be further considered. In this study, immune responses induced by booster vaccines in mice were compared, including responses by inactivated vaccines, mRNA vaccines based on the WT strain, and mRNA vaccines based on B1.1.529, one of the VOCs that has caused the most serious harm. For those who have received two-dose full-schedule immunization with alum-adjuvanted inactivated SARS-CoV-2 vaccines, this study provides a vaccine booster strategy that corresponds to mRNA vaccines based on variant viruses, but not of the WT strain, which could be the best choice to combat VOCs.

## 2. Materials and Methods

### 2.1. Vaccines

The inactivated SAR-CoV-2 vaccine KEWEIFU^TM^, which has been used in a phase II clinical trial, was supplied by the Institute of Medical Biology, Chinese Academy of Medical Sciences (IMBCAMS). KEWEIFU^TM^ contains the prototypic SARS-CoV-2 strain (also known as KMS-1, GenBank accession number MT226610.1) double inactivated with formaldehyde plus β-propiolactone and adjuvanted with aluminum hydroxide [10].

LNP-mRNA vaccines were prepared by a modified procedure as described previously [11]. Briefly, DNA sequences encoding the whole length of the SARS-CoV-2 S protein for the WT and B1.1.529 strains were synthesized with 5′ and 3′ untranslated regions (Sangon Biotech Co., Ltd., Shanghai, China). mRNAs were subsequently produced and purified by an mRNA synthesis kit (APEXBIO Technology, Houston, TX, USA) and Monarch^®^ RNA purification columns (NEW ENGLAND BioLabs Inc., Ipswich, MA, USA). The mRNA products were mixed with lipids (dissolved in ethanol at molar ratios of 50:10:37.5:2.5 (MC3:DSPC:cholesterol:DMG-PEG 2000)) at a ratio of 3:1 with a microfluidic mixer (Precision Nanosystems, Inc., Vancouver, BC, Canada). Formulations were dialyzed against PBS in centrifugal filtration tubes (Millipore, Tullagreen, Carrigtwohill, Co., Cork, Ireland), passed through a 0.22 µm syringe filter, and stored at −80 °C until use. LNP particle sizes were determined by a Zetasizer Nano ZS particle size analyzer (Malvern Panalytical, Malvern, UK), 1% denatured agarose gel, and Quant-it^TM^ RiboGreen RNA Assay Kit (Thermo Fisher, Eugene, OR, USA). The encapsulation efficiency was calculated as described elsewhere [11].

### 2.2. Mouse Studies

Specific pathogen-free female BALB/C mice (6–8 weeks, 20–22 g) were supplied by the Central Animal Services of the IMBCAMS and maintained under standard pathogen-free conditions before immunogenicity studies. Ethical approval for animal research was provided by the Institutional Animal Care and Use Committee (IACUC) of the IMBCAMS (access number DWSP202207015).

For inactivated vaccines, mice were intramuscularly administered 1/10 of the human immunogen dose, which consists of 15 enzyme-linked immunosorbent assay units (EU) of viral antigen and 25 μg of aluminum hydroxide. For mRNA vaccines, mice were intramuscularly injected with 14 μg mRNA/50 μL for each dose. Mice were sacrificed 2 weeks after the final immunization, and whole-blood samples were collected via cardiac puncture. After centrifugation at 1000× *g* for 30 min, serum was obtained and stored at −80 °C until use.

### 2.3. IgG/IgG1/IgG2a Titer Measurement

The levels of spike protein-specific antibodies in serum samples of immunized mice collected at different points were determined by enzyme-linked immunosorbent assay (ELISA). Recombinant S-trimer 6P protein, the S protein of the SARS-CoV-2 WT strain, was purchased from Atagenix (Wuhan, China). SARS-CoV-2 (B1.1.529) S protein, tagged with His at the C-terminus, was obtained from Vazyme (Nanjing, China). S proteins were precoated in 96-well microplates at a final concentration of 2 μg/mL, and the plates were incubated overnight at 4 °C. After being washed three times with phosphate-buffered saline containing Tween-20 (0.05% *v/v*) (PBST), microplates were blocked with 5% (*w/v*) skim milk in PBS for 1 h. Serial dilutions of mouse sera were added, and the plates were incubated for another 3 h at room temperature. Goat anti-mouse IgG (1:10,000), IgG1 (1:500), and IgG2a (1:2000) conjugated with horseradish peroxidase (HRP) were used as detection antibodies. IgG/IgG1/IgG2a titers were defined as the endpoint dilutions showing cut-off signals above OD450 = 0.15, and antibody titers lower than 50 at a dilution of 1:500 were defined as 50 for calculations [9].

### 2.4. Enzyme-Linked Immunospot (ELISPOT) Assay

The frequencies of IL-2 and IFN-γ secretion by splenocytes were detected by an ELISPOT assay kit (BD) according to the manufacturer’s protocol. Briefly, splenocytes were seeded in 96-well assay plates precoated with 2 μg/mL antibodies against IL-2 and IFN-γ at a final concentration of 3 × 10^5^ cells/well. Recombinant S-trimer 6P protein at 20 µg/mL was used to stimulate S protein-specific T-cell responses, and the same volume of a solution of PMA+ ionomycin was used as a positive control. Spots were counted with an ELISPOT reader system (Autoimmun Diagnostika GmbH, Strassberg, Germany) after immunoimaging [12].

### 2.5. Pseudovirus-Based Neutralization Assay

SARS-CoV-2-Fluc pseudotyped WT and B.1.1.529 viruses, provided by Vazyme (Nanjing, China), were used for neutralization assays in biosafety level 2 facilities. Specifically, the pseudoviruses were diluted to a 2 × 10^4^ 50% tissue culture infectious dose (TCID50)/mL with complete DMEM (DMEM supplemented with 10% FBS and 1% penicillin-streptomycin), and sera were serially threefold diluted. Diluted SARS-CoV-2 pseudoviruses (50 μL) and serial dilutions of immune serum (50 μL) were co-incubated for 60 min at 37 °C with 5% CO_2_. Subsequently, HEK293-ACE2 cells (2 × 10^5^ cells/50 μL/well) were seeded in each mixture for another 48 h. A cell control (CC), in which only cells with culture medium were added, and a virus control (VC), in which only pseudoviruses and cells but no serum were added, were established separately. According to the manual protocol, luciferase activity was assessed by a microplate reader (Tecan, Spark), and the nAb titers were calculated based on serum inhibiting 50% of pseudoviruses at the diluted concentration.

### 2.6. Statistical Analysis

Data were analyzed with GraphPad Prism 9.2( GraphPad Software Inc., La Jolla, CA, USA) and are shown as the mean ± standard deviation (SD). Significant differences among experimental groups were analyzed by t-tests and one-way analysis of variance followed by Dunnett’s multiple comparisons tests. Asterisks represent the p-value classification: * *p* < 0.05; ** *p* < 0.01, and *** *p* < 0.001.

## 3. Results

### 3.1. LNP-mRNA Vaccines Were Formulated with Efficient Encapsulation of mRNA with a Uniform Particle Size

As shown in Figure 1A, after immunization with two doses of inactivated SARS-CoV-2 vaccine, another two doses of prototypic inactivated vaccines or mRNA vaccines encoding the S protein of the WT and B.1.1.529 strains were administered as boosters (abbreviated B), and the groups were designated B-inactivated, B-mRNA WT, and B-mRNA omicron, respectively. Groups that were immunized with two doses of mRNA vaccines for the WT and B.1.1.529 strains with the same schedules but were not primed with two doses of inactivated vaccines were set as the control mRNA vaccine groups (mRNA WT and mRNA omicron groups). Mice immunized with the same volume of PBS were included as the control group.

mRNA encoding various viral surface S antigens was formulated as LNP-encapsulated mRNA vaccines for each group, with a mass of 170 μg (approximately 14 μg per dose, excluding natural waste). The encapsulation efficiency of nucleic acids was approximately 100%, as determined by an RNA Assay Kit (Figure 1B). The LNP was on average 80 nm in diameter, regardless of whether the vaccine was based on the WT or B.1.1.529 strain (Figure 1C). Determination of the polydispersity index (PDI), a measure of the heterogeneity of a sample based on size, revealed that the LNP particles had good uniformity, with a mean value of 0.06 for the mRNA-WT vaccine and an average value of 0.10 for the mRNA omicron vaccine (Figure 1D).

### 3.2. After Two Doses of an Inactivated SARS-CoV-2 Vaccine, Boosting with the mRNA Vaccine Induces the Elevation of IgG Titers

For the S protein of the WT strain (Figure 2A), two booster doses of mRNA vaccine (group B-mRNA WT and group B-mRNA omicron) induced the production of higher IgG titers, with mean IgG titers of 725,333 and 469,333, respectively. Two doses of the WT mRNA vaccine resulted in the same average IgG titers as those in the mRNA vaccine-boosted groups (mRNA WT group vs. B-mRNA WT group/B-mRNA omicron group = ns). Regarding the responses to the S protein of the B.1.1.529 Group (Figure 2D), the IgG titers of mRNA-boosted groups (mean IgG titers of the B-mRNA WT and B-mRNA omicron group of 256,000 and 160,000, respectively) were higher than those of the inactivated group (mean IgG titers of 69,333) and two-dose mRNA vaccine-immunized groups (mean IgG titers for the mRNA WT and mRNA omicron group of 80,000 and 122,675, respectively). IgG1 and IgG2a titers in mouse sera against the S protein of the WT strain (Figure 2B,E) and B.1.1.529 strain (Figure 2C,F) were measured, and it was observed that the trends between groups for IgG1 and IgG2a were similar to those for IgG.

### 3.3. Regarding Two-Dose Inactivated Vaccine Immunization, Boosting with Two Doses of SARS-CoV-2 mRNA Vaccines Induced Potent Cell-Mediated Immunity

After stimulation with SARS-CoV-2 S-trimer 6P protein, the secretion frequencies of secretion of IL-2 and IFN-γ, two important cytokine indicators of Th1 orientation, among splenocytes of immunized mice were analyzed by ELISPOT. The frequencies of IL-2 (Figure 3A) and IFN-γ (Figure 3C) secretion in the mRNA vaccine-immunized groups, regardless of whether they were boosted with mRNA vaccines or immunized with mRNA vaccines directly, were significantly higher than those in the B-inactivated group, in which all four doses were injected with only inactivated vaccines. The trends of IL-2 (Figure 3B) and IFN-γ (Figure 3D) secretion were consistent. The mean numbers of spots, regardless of IL-2 or IFN-γ, for the B-mRNA WT group (IL-2/IFN-γ: 581/470) and the mRNA WT group (IL-2/IFN-γ: 553/574) were significantly higher than that for the B-mRNA omicron group (IL-2/IFN-γ: 421/273) and mRNA omicron group (IL-2/IFN-γ: 468/248), suggesting that the mRNA vaccine based on the WT strain could induce more potent cell-mediated immunity (CMI) than the one based on the B.1.1.529 strain. In summary, mRNA vaccines induced a more robust CMI response than inactivated vaccines, which was consistent with previous reports [13]. After prime injection with two doses of inactivated vaccines, the CMI response could be enhanced to some extent.

### 3.4. The Cross-Protection Afforded by Neutralizing Antibodies (nAbs) Produced in Response to SARS-CoV-2 Vaccines Was Reduced

When mixed with WT strain SARS-CoV-2 pseudoviruses, serum from the B-inactivated/B-mRNA WT/B-mRNA omicron/mRNA WT groups showed obvious neutralization activity, with mean nAb titers of 3282/10,525/4405/1119 (Figure 4A). Unsurprisingly, serum from the B-mRNA WT group exhibited the best neutralization effect, as the activity of 50% of pseudoviruses was inhibited when sera were diluted 10,525-fold (Figure 4A, B-mRNA WT group). Nonetheless, for the B-mRNA omicron and mRNA omicron groups, with moderate and low activity against the WT strain, outstanding neutralizing activity against the B.1.1.529 strain was observed, with mean nAb titers of 1796 and 2578 (Figure 4B, B-mRNA omicron and mRNA omicron groups).

## 4. Discussion

Due to the general susceptibility of humans to SARS-CoV-2, vaccination is still the most economical and effective public health intervention for pandemic prevention. Since it was first reported in November 2021, the omicron (B1.1.529) variant has spread quickly and infected millions of people, as viral infectivity has increased 13-fold [14] and vaccine effectiveness has decreased [15]. However, the extremely high rate of omicron infection did not result in more severe disease in the real world, which is most likely due to widespread vaccination [16,17]. It was indicated that vaccination with the complete vaccination schedule plays an important role in pandemic prevention and control, mainly reducing the severity and mortality of COVID-19, regardless of infection with the prototypic strains or VOC strains [8,18].

On 26 January 2023, the FDA proposed that most Americans receive one annual COVID-19 vaccine, similar to the recommendation for influenza, meaning that considering which vaccines to use for boosters will be a long-standing issue. The groups immunized with two-dose alum-adjuvanted inactivated SARS-CoV-2 vaccines develop relatively diminished antibody titers and uncertain cellular immunity compared with those individuals immunized with other forms of SARS-CoV-2 vaccines, such as mRNA/adenovirus vaccines, which leads to questions about which booster vaccines should be taken in the context of numerous emerging mutant strains and gradually decreasing antibody titers [19,20]. Studies have shown that heterologous vaccination is a promising strategy to maximize vaccine immunogenicity [21], and whether mRNA SARS-CoV-2 vaccines are a good choice for inactivated SARS-CoV-2 vaccine-vaccinated individuals is discussed in this study.

Based on our previous report [9], the first three-dose schedules were set as 0–28–84 days in this study (Figure 1A, B-inactivated/B-mRNA WT/B-mRNA omicron groups). The immunization schedule of mRNA vaccines was further determined based on a 28-day interval according to our past studies on mRNA vaccines [11] (Figure 1A, mRNA-WT/mRNA-omicron groups). The mRNA encapsulation efficiency of LNPs was approximately 100% in this study, which is consistent with our laboratory’s experience that the mRNA encapsulation efficiency of LNPs was superior to that for DNA, with a rate of 100% for mRNA but 70% for CpG oligodeoxynucleotides (ODNs) [11,12,22]. The structural mechanisms are still largely unclear, which may be dependent on the sequence and electric charge of nucleic acids. As a cationic liposome delivery system, LNPs guarantee that mRNA can be encapsulated in the internal cavity, which can improve the stability of mRNA in vivo and facilitate the function of mRNA [2].

mRNA vaccines have been reported to stimulate the production of potent antibodies, making them a favorable choice for boosters against the current SARS-CoV-2 VOCs [23,24,25]. Our study revealed that compared with inactivated vaccines, the mRNA vaccines boosted humoral immune responses and CMI to some extent. Initially, IgG antibody titers increased 4–6 times and 2–4 times in response to the WT and B1.1.529 strains, respectively (Figure 2A,D). The IgG1 and IgG2a titers showed the same increase as the IgG titers (Figure 2B,E, Figure 2C,F). According to the IgG1/IgG2a ratio (Appendix A), an indicator of Th1-Th2 responses, the mRNA vaccines surprisingly induced a more Th2-biased immune response in this study. mRNA vaccines are more frequently reported to induce a Th1 immune response [26], and the different techniques and constituents of LNPs may be one of the reasons for this distinctive result.

In addition to their stimulation of humoral immunity, it has been demonstrated that mRNA-LNPs can elicit polyfunctional, antigen-specific, CD4^+^ T-cell responses and potent neutralizing antibody responses. Given that a full immunization schedule of inactivated vaccines rarely stimulates a T-cell response, whether mRNA vaccines administered as boosters could induce robust CMI was analyzed in our study. In clinical trials, induced CMI was strengthened by 1.5-fold when BNT162b2, an mRNA vaccine, was inoculated as the third booster vaccine for two-dose inactivated SARS-CoV-2 vaccine-immunized individuals [27]. The ELISPOT results showed that the mRNA vaccines we made in our laboratory also enhanced CMI by approximately 4-fold (IL-2, Figure 3B) and 8-fold (IFN-γ, Figure 3D).

Several studies have reported that breakthrough infections occurred in some vaccine recipients, indicating that there is a strong correlation between immune protection and the titers of nAbs [28,29]. In our study, we found that high levels of nAbs against the WT virus were boosted by mRNA vaccines, with a value of 10,525/4405 in the B-mRNA WT/B-mRNA omicron group (Figure 4A). Compared with the B-inactivated group, nAbs against the B.1.1.529 strain also reached 1796 through boosting with the mRNA omicron vaccine, but did not become significantly higher through boosting with the mRNA WT vaccine (Figure 4B). What surprised us was that immunized mRNA omicron vaccines alone (mean nAb titers of the mRNA omicron group: 2578) achieved slightly higher nAbs than priming with two doses of inactivated vaccines (mean nAb titers of the B-mRNA omicron group: 1796). It seems like inactivated vaccines, which are based on the WT strain, decrease the nAbs against the B.1.1.529 strain of the mRNA omicron vaccine. Together with the results of our previous study, this finding indicates that the vaccine strain in the first dose is critical for the nature of the induced nAbs [30]. Accordingly, we suggest that individuals who have received two doses of inactivated vaccines should be boosted with vaccines for variant strains to produce nAbs corresponding to variant strains, regardless of whether the vaccines are mRNA vaccines or other kinds of vaccines.

As of today, new vaccines against variants are being developed with the appearance of more variants. Moderna has developed bivalent mRNA vaccines, including mRNA-1273.211 and mRNA-1273.214, to combat the WT strain and delta and omicron variants, respectively. The protective effect of new vaccines still needs long-term market testing, at least regarding the concerns with rapidly developing vaccines, such as the antibody enhancement effect and thrombosis [29,30]. At present, this study provides the optimal vaccination strategy for existing vaccines with proven safety against SARS-CoV-2 variants. For the enormous number of individuals that have been immunized with two doses of inactivated vaccines, more studies are needed to provide scientifically informed guidance on future steps.

## 5. Conclusions

In conclusion, based on initial vaccination with two inactivated vaccines, boosting with mRNA vaccines can obtain better humoral and cellular immunity, regardless of whether the WT or omicron strain is used, showing that mRNA SARS-CoV-2 vaccines are a better choice for boosting than inactivated vaccines. Analysis of the nAb titers showed that for those who have not yet received any vaccine, vaccination with the corresponding vaccine could be the best choice to defend against the corresponding strain. For those who have been immunized with two doses of inactivated vaccines, mRNA vaccines based on variant strains could be an option to provide dual protection against the WT strain and variants.

## Figures and Tables

**Figure 1 viruses-15-01351-f001:**
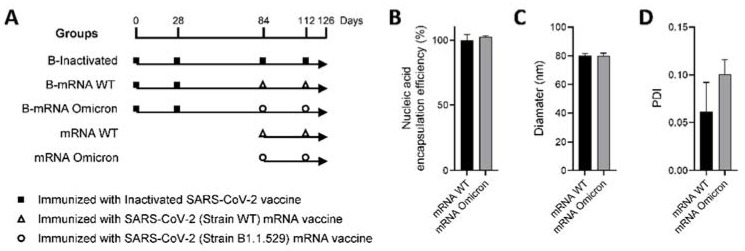
Immunization schedule and characterization of the mRNA vaccines. (**A**) Illustrated immunization schedules of the groups. The solid box, hollow triangle, and hollow circle represent immunization with the inactivated SARS-CoV-2 vaccine (KEWEIFU^TM^) and mRNA vaccines encoding the S protein of the WT and B1.1.529 strains, respectively. (**B**) The nucleic acid encapsulation efficiency of mRNA-LNP vaccines. (**C**,**D**), The diameter and polydispersity index (PDI) of mRNA-LNP vaccines were measured by a Zetasizer Nano ZS particle size analyzer, and each test was repeated three times.

**Figure 2 viruses-15-01351-f002:**
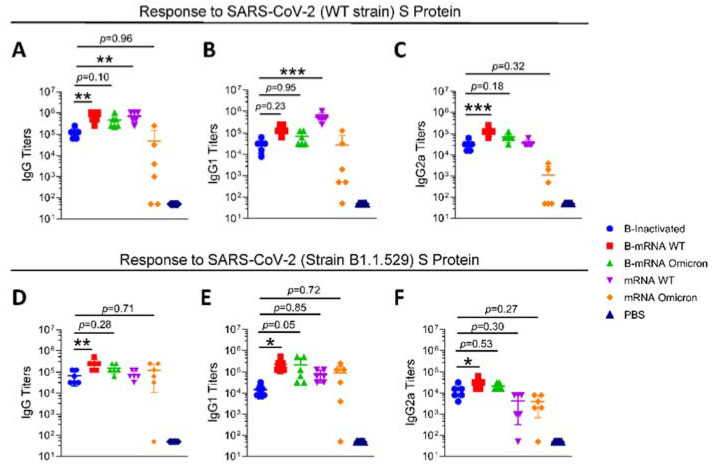
Antibody titers detected by enzyme-linked immunosorbent assay (ELISA). S proteins of the WT strain (**A**–**C**) and B1.1.529 strain (**D**–**F**) were precoated onto microplates, and the IgG (**A**,**D**), IgG1 (**B**,**E**), and IgG2a (**C**,**F**) levels in sera from different groups were measured. N = 6, points represent individual mice. Data were compared using one-way analysis of variance followed by Dunnett’s multiple comparisons tests, with the inactivated group as the control. * *p* < 0.05, ** *p* < 0.01, *** *p* < 0.001.

**Figure 3 viruses-15-01351-f003:**
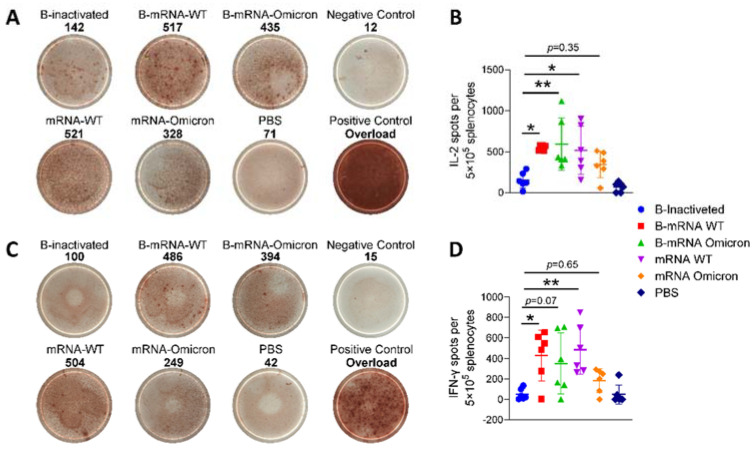
Frequencies of IL-2 and IFN-γ secretion by isolated splenocytes of mice were assayed by enzyme-linked immunospot (ELISPOT). Representative images of captured ELISPOT spots are shown in the left panels (**A**,**C**), the value of each representative image was marked at the top of the pictures. The number in the Positive Control group was too high for the ELISPOT reader system to detect, thus no specific value was shown. Numbers of IL-2-secreting (**B**) and IFN-γ-secreting (**D**) cells were counted. N = 6, points represent individual mice. Data were compared using one-way analysis of variance followed by Dunnett’s multiple comparisons tests, with the inactivated group as the control. * *p* < 0.05, ** *p* < 0.01.

**Figure 4 viruses-15-01351-f004:**
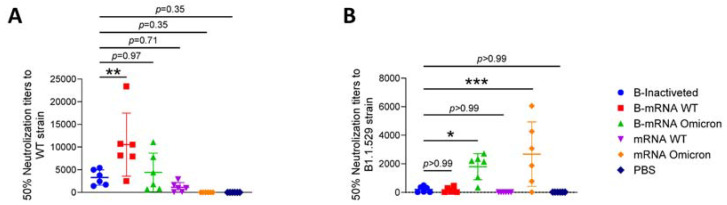
SARS-CoV-2 neutralization (nAb) titers in serum from immunized mice. (**A**) nAb titers in sera produced in response to the WT strain pseudovirus. (**B**) nAb titers in sera produced in response to the B1.1.529 strain pseudovirus. Each experiment was repeated twice, and the mean values were used for statistical analysis. N = 6, points represent individual mice. Data were compared using one-way analysis of variance followed by Dunnett’s multiple comparisons tests, with the inactivated group as the control. * *p* < 0.05, ** *p* < 0.01, *** *p* < 0.001.

## Data Availability

All data used in this study are available from the corresponding author by request.

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
