# Peer review of "Comparison of Immune Responses between Inactivated and mRNA SARS-CoV-2 Vaccines Used for a Booster Dose in Mice"

_viruses, 2023, doi:10.3390/v15061351_

Round 1
Reviewer 1 Report
Dear Editor/ Authors
Authors have studied the immune effects of mRNA vaccines based on the WT (prototypic strain) and omicron (B1.1.529) strains for use as booster vaccines in mice. The authors documented that mice who have received two doses of inactivated vaccines should be boosted with vaccines for variant strains to produce nAbs corresponding to variant strains, regardless of whether the vaccines are mRNA vaccines or other kinds of vaccines. Finally, this study provides the optimal vac-cination strategy for existing vaccines with proven safety against SARS-CoV-2 variants. In conclusions; boosting with mRNA vaccines can obtain better humoral and cellular immunity, regardless of whether the WT or omicron strain is used, showing that mRNA SARS-CoV-2 vaccines are a better choice for boosting than inactivated vaccines. Secondly, mRNA vaccines based on variant strains could be an option to provide dual protection against the WT strain and variants of concern.
The topic is very intriguing, and in my opinion evaluated manuscript is suitable for publication in Viruses journal in this shape. I'm most impressed with the high quality of presented paper. In my opinion all aspects of presented paper are well documented.
Author Response
Many thanks for your review. Hope our work not only meet the topic of Virus journal, but also provide some guidance for the control of SARS-CoV-2.
Reviewer 2 Report
Research into the development of vaccines for the prevention of COVID-19 is still important and needed. This is especially true of vaccines that can protect against newly emerging variants of this virus. In their work, Ning Luan et al. are engaged in research in this area.
The article is written at a good scientific level. The authors use a number of modern methods to assess immunity after immunization. The article has a sufficient number of illustrations. Statistical processing of the results was carried out correctly.
Author Response
Thank you for your valueble review.
Reviewer 3 Report
The authors described vaccination strategies against SARS-CoV-2 in mice. Comparisons are made between vaccination schedules involving SARS-CoV-2 inactivated, mRNA WT and mRNA Omicron vaccines. While only a small cohort of 6 mice are used per vaccine arm, the trends are clear and align with known literature. Overall, the manuscript is well written, and results are clearly described. The manuscript is good for publication once the following comments are addressed:
Figure 3A, C: It would be good to add the numerical number of spots present for each representative image, to allow the reader to better appreciate the differences.
Figure 4: The authors should decide if they are showing the p-values for all comparisons or just for comparisons that are significant or trending significance, and keep it constant between both graphs. It is also unclear why only the neutralization graph for B1.1.529, but not that for the WT, has a negative axis.
Discussion: (Line 279) "According to the IgG1/IgG2a ratio (data not shown), an indicator of Th1-Th2 responses, the mRNA vaccines surprisingly induced a more Th2-biased immune response in this study" If the authors intend to discuss about T-helper cell responses, data such as IgG1/IgG2a ratio should be shown, either in the main or supplementary figures (currently this data is not shown).
It isn't clear to this reviewer that (Line 293) "Similar trends were observed in the flow cytometry analysis, regardless of the CD4+ T-cell response (Supplementary Figure 1) and CD8+ T-cell response (Supplementary Figure 2), although there were no statistically significant differences." In some instances it even appears that the inactivated booster (blue) worked better than the mRNA boosters. The authors should consider revising or removing this statement.
(Line 298) " In our study, we found that high levels of nAbs against WT virus were boosted by mRNA vaccines (Figure 4A), but measurement of nAb titres against the omicron variant did not imply the same trend (Figure 4B). Together with the results of our previous study, this finding indicates that the vaccine strain in the first dose is critical for the nature of the induced nAbs [30]." The authors should be more specific about the comparisons made within the statements, especially the "mRNA vaccines" being discussed. The mRNA Omicron vaccine worked well in providing neutralizing responses to the B.1.1.529 strain, regardless if it was given as a mRNA booster after 2 x inactivated WT vaccines, or as 2 primary vaccine doses.
Minor comment: Figure S1, S2: While the authors used whole spike trimer protein as a stimulus here, similar to that used in the ELISPOT, both experiments may have benefited if a SARS-CoV-2 spike or RBD peptide pool was used instead for better stimulation of SAR-CoV-2 specific T cells.
Author Response
I'm honored to have your review. I have changed the minors you mentioned in the revised manuscript as bellows:
- I have added the numerical of spots present for each representative image in Figure 3A, 3C. Revised Figure 3 has been attached in the line 202 in the revised manuscript, and figure legend also be re-descripted in the line 205-207 correspondingly.
- Thank you for your mention that I haven’t noticed that I used the different analysis method in the Figure 4A and Figure 4B. I have re-analyzed data in Figure 4A and 4B, using one‐way analysis of variance followed by Dunnett's multiple comparisons tests, with the inactivated group as the control. In the revised Figure 4, which has been attached in the line 238 of revised manuscript, p-values for all comparisons were shown, including comparisons that are significant or trending significance, negative axis in Figure 4B also been modified by change the Y axis range. Redrawn Figure 4 didn’t affect results we indicated in the manuscript.
- I have attached the results of IgG1/IgG2a ratio in supplementary Figure 1, and changed the word parts in line 286 of revised manuscript.
- Thanks for your valuable opinion, I have removed the statement about the flow cytometry analysis (line 300 in revised manuscript), and deleted the corresponding figures in supplementary material. We indeed haven’t noticed the importance of stimulation proteins for the results of ELISPOT and flow cytometry. Depending on your kindly reminder, I deleted the unclear description about the results of flow cytometry to avoid the misunderstanding of whole results, and we would comprehensively consider the influence of stimulus in the future study to avoid presenting uncertain results.
- I have made more specific statements about the comparisons made within the mRNA vaccines groups; details have been reworked in line 304-312 of revised manuscript. Thank you for your thoughtful mention, which let me realized that what I described will disturb readers to understand our results.